# PbF$_2$–CdF$_2$–SrF$_2$ Ternary Solid Solution: Crystal Growth and Investigation

Irina I. Buchinskaya *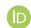, Ivan O. Goryachuk, Nikolay I. Sorokin, Victor I. Sokolov and Denis N. Karimov

Shubnikov Institute of Crystallography, Federal Scientific Research Centre "Crystallography and Photonics", Russian Academy of Sciences, Leninsky Prospekt 59, Moscow 119333, Russia

* Correspondence: buchinskayaii@gmail.com

**Abstract:** Crystals based on alkaline earth metal difluorides are widely used optical materials. In this study, in order to expand the range of optical matrices, multicomponent Pb$_{1-x-y}$Cd$_x$Sr$_y$F$_2$ (0.27 < x < 0.55, 0.06 < y < 0.18) solid solution crystals with a fluorite structure (sp. gr. *Fm-3m*) were grown from melt using the vertical directional crystallization technique for the first time. The densities and refractive indices of the grown crystals vary depending on the quantitative content components (*x* and *y*) in the ranges of 6.6039(5)–7.5232(5) g/cm$^3$ and 1.6403–1.7084, respectively. The optical transmission and electrochemical impedance spectra were studied. The homogenous composition regions of non-cellular crystallization of this ternary solid solution at a crystallization rate of 6 mm/h and an interface temperature gradient of 80 deg/cm were experimentally determined as 0.30 < x < 0.35, 0 < y < 0.6. These grown crystalline materials may be of interest as high-density highly refractive cubic isomorphic hosts and low-temperature ionic conductors (~2 × 10$^{-5}$ S/cm at room temperature) for various applications.

**Keywords:** multicomponent crystals; solid solutions; alkaline earth metal fluorides; PbF$_2$; CdF$_2$; SrF$_2$; directional crystallization; optical materials; high density; refractive index; phase diagram; ionic conductivity; electrochemical impedance





## 1. Introduction

Difluorides *M*F$_2$ (*M* = Ca, Sr, Ba, Cd and Pb), which belong to the fluorite (CaF$_2$) structural type, sp. gr. *Fm-3m*, Z = 4, are important multifunctional optically isotropic materials. The diversity of *M*F$_2$ fluorite matrices can be significantly increased by the crystallization of isostructural solid solutions *M*$_{1-x}$*M'*$_x$F$_2$, where *M* ≠ *M'* are divalent cations. The transition from simple one-component to multicomponent systems, that is, the synthesis of mixed solid solutions with a certain type of crystal structure, is an effective method for creating new crystalline materials with the required and finely tuned properties for practical applications.

Variants of congruent melting of compositions and, accordingly, solidification of melts, for which the distribution coefficients of all components are equal to unity, are of particular importance for growing homogeneous multicomponent single crystals. Continuous solid solutions *M*$_{1-x}$*M'*$_x$F$_2$ of a fluorite structure with a temperature minimum point on the liquidus and solidus lines are formed in the *M*F$_2$–*M'*F$_2$ systems with such combinations of cations as Ca-Sr, Ba-Sr, Cd-Sr, and Pb-Cd: CaF$_2$–SrF$_2$ [1], BaF$_2$–SrF$_2$ [2], CdF$_2$–SrF$_2$ [3], and PbF$_2$–CdF$_2$ [3,4]. The solid solution corresponding to the composition of the temperature minimum melts congruently, which ensures the growth of homogeneous crystals of such solid solutions in the vicinity of these compositions. Such crystals were grown previously and studied to a limited extent and are listed in Table 1.

**Table 1.** Data on selected mixed crystals based on alkaline earth difluorides.

| The Compositions of Crystals Grown in the $MF_2–M'F_2$ and $MF_2–M'F_2–M''F_2$ Systems | Melting Point, K | Literature Sources |
|---|---|---|
| $Ca_{0.582}Sr_{0.418}F_2$ | 1646 | [1,5] |
| $Ca_{0.59}Sr_{0.41}F_2$ | - | [6,7] |
| $Ca_{0.5}Sr_{0.5}F_2$ | - | [8] |
| $Ca_{1-x}Sr_xF_2$ ($0 \leq x \leq 1$) $Ca_{0.257}Sr_{0.743}F_2$—studied composition | - | [9,10] |
| $Ba_{0.66}Sr_{0.34}F_2$ | 1850 | [11] |
| $Cd_{0.75}Sr_{0.25}F_2$ | 1298 | [12] |
| $Cd_{0.77}Sr_{0.23}F_2$ | - | [7,13] |
| $Pb_{0.67}Cd_{0.33}F_2$ | $1023 \pm 5$ | [7] |
| $Ca_{0.3(3)}Sr_{0.3(3)}Ba_{0.3(3)}F_2$ | 1423–1483 | [14,15] |

It should be noted that the $CaF_2–SrF_2$ system is the most designed in terms of growing crystals and the most practically significant until now. Rare-earth-doped $Ca_{1-x}Sr_xF_2$ solid solution crystals (and optical ceramics based on it) were grown and studied in [16–22] for various photonic applications. Other examples of utilizations of solid solutions of difluorides are also known. For example, $Cd_{1-x}Pb_xF_2$ crystals can be used to construct reversible electrochemical cells [23], $PbF_2$, $CdF_2$ and solid solutions based on it, both undoped and doped with active ions—as scintillator crystals [24] and film optical coatings [25,26].

The number of chemical compositions suitable for crystallization from a melt of homogeneous two-component hosts is also limited. An increase in the number of components in a solid solution and a transition to triple or more component combinations can be of great interest. At the same time, there is the problem of the heterogeneity of the chemical composition and, as a consequence, the physicochemical properties of such objects getting sharper. Therefore, it is important to search for multicomponent compositions that are close to congruent.

The theoretical foundations of this approach were previously considered in detail [27]. In the ternary systems, $MF_2–M'F_2–RF_3$, where a temperature minimum is realized on the melting curves in the bounding binary $MF_2–M'F_2$ system and maxima in the $MF_2–RF_3$ and $M'F_2–RF_3$ ($R$—rare earth elements, REEs), congruent saddle-type points are experimentally discovered, but for the maxima and saddles, the melt crystallization process is unstable. Only the minimum points on the melting surfaces of solid solutions are stable from the point of view of the crystallization process. But such a combination of three or more difluorides with a fluorite structure has not been found to date. Therefore, the search for multicomponent compositions with compositions close to congruent melting remains actual and is crucial for the development of inorganic materials science. Recently, single crystals of a ternary solid solution in the $CaF_2–SrF_2–BaF_2$ system were grown and studied in [14,15] in the range of compositions close to $Ca_{0.3(3)}Sr_{0.3(3)}Ba_{0.3(3)}F_2$ (CaSrBaF$_6$) with a high degree of uniformity in the axial distribution of components. Apparently, the nature of their crystallization is close to congruent. Crystals based on the $Ca_{1-x-y}Sr_xBa_yF_2$ solid solution were previously considered as a unique material for photolithography in extreme ultraviolet [28].

The ternary system $PbF_2–CdF_2–SrF_2$ is a system that is attractive for research because it consists of a continuous field of a fluorite solid solution and the two $PbF_2–CdF_2$ and $CdF_2–SrF_2$ systems that bind it, which contain compositions of congruent melting points of temperature minima.

As part of the search for new functional high-density, isotropic, and highly refractive materials, the purpose of this study is to analyze the liquidus topology of the $PbF_2–CdF_2–SrF_2$ system, grow the ternary solid solution crystals from melt, and investigate their homogeneity and optical and electrical properties. It is known that materials based on the $PbF_2–CdF_2$ system are fluoride superionic conductors due to high conductivity ($\sigma$) [29–31]. The electrical conductivity maximum corresponds to the $Pb_{0.67}Cd_{0.33}F_2$ composition of

the temperature minimum point in the phase diagram $PbF_2$–$CdF_2$. Previously, the effect of isomorphic substitutions of aliovalent ($Li^+$, $Na^+$, $Ce^{3+}$ [32]) and isovalent ($Mn^{2+}$ [29]) impurities on the ionic conductivity of $Pb_{0.67}Cd_{0.33}F_2$ was studied. In the case of aliovalent admixtures, no increase in conductivity values was observed, and in the case of an isovalent impurity, the value of $\sigma$ increased by ~3 times. Therefore, measurements of the electrical properties of some multicomponent compositions $(Pb_{0.67}Cd_{0.33})_{1-y}Sr_yF_2$ in the $PbF_2$–$CdF_2$–$SrF_2$ system will be carried out to evaluate the potential of these objects for application in solid-state ionics.

## 2. Materials and Methods

Crystals were grown from the melt using vertical directional crystallization in a two-zone resistive furnace in a graphite heating unit in a fluorinating atmosphere (He/$CF_4$ mixture). Commercial powders of $PbF_2$ (99.99%, Chemcraft Ltd., Kaliningrad, Russia), $CdF_2$ (99.9%, Chemcraft Ltd., Kaliningrad, Russia), and $SrF_2$ (99.995%, Lanhit Ltd., Moscow, Russia) preliminarily dried in a vacuum for 2 h at a temperature of 400 K and then remelted and purified using directional crystallization in a fluorinating atmosphere were applied as starting reagents. The temperature gradient in the growth zone was ~80 deg/cm, the crucible pulling rate was ~6 mm/h, and the crystal cooling rate was 100 deg/h. Open-type multi-cell graphite crucibles were used in the crystallization process.

X-ray diffraction (XRD) analysis of the crystals was performed on a Rigaku MiniFlex 600 X-ray powder diffractometer ($CuK_\alpha$ radiation). Registration of diffraction patterns was carried out in the range of angles 2θ from 10° to 120°. Calculations of unit cell parameters within sp. gr. *Fm-3m* were performed with Le Bail full-profile analysis using the JANA2006 software package [33].

The elemental composition of samples cut from crystal boules was controlled by X-ray fluorescence analysis (XRFA) on an Orbis microanalyzer (EDAX).

The crystal refractive indices $n$ were measured at a wavelength of $\lambda = 632.8$ nm at $T = 293$ K on a Metricon 2010/M prism refractometer (Metricon Corporation). The measurement technique is based on determining the critical angle of incidence at which light begins to pass into the sample volume through the surface of the measuring prism (similar to the Abbe refractometer). The refractive indices were determined with an accuracy of no worse than $\pm 0.0005$ from the dependence of the reflection coefficient $R$ of the radiation on the angle of incidence θ under conditions of frustrated total internal reflection at the TE polarization of the incident He-Ne laser beam using the Snell–Descartes formula:

$$n = N \sin (\theta_{crit}), \tag{1}$$

where $N = 2.15675$ is the refractive index of the measuring prism of the device, and $\theta_{crit}$ is the critical angle of total internal reflection.

The transmission spectra of the crystals were recorded using a Cary 5000 spectrophotometer (Agilent Technologies, California, CA, USA) and an FTIR-8100 IR-Fourier spectrometer (Shimadzu) in the wavelength range $\lambda = 0.2$–15 μm.

The homogeneity of the samples was studied using a POLAM L-213M optical microscope (Russia).

The direct current electrical conductivity of one of the grown crystals was measured with impedance spectroscopy at the temperature range of 293–406 K. The sample was a plane-parallel plate 1.5 mm thick, on the working surfaces of which inert electrodes (Leitsilber silver paste) were deposited. The area of electrical contacts was 25 $mm^2$. Measurements of the impedance $Z^*(\omega)$ of the electrochemical system Ag | crystal | Ag were performed in the frequency range 5–5 × $10^5$ Hz and at a resistance of 1–$10^7$ Ω (Tesla BM-507 impedance meter). The relative measurement error $Z^*(\omega)$ was 5%. The impedance measurement technique is described in detail in [34].

## 3. Results and Discussion

### 3.1. Analysis of the $PbF_2$–$CdF_2$–$SrF_2$ System—Search for a Region of Homogeneous Compositions

As mentioned above, the $PbF_2$–$CdF_2$–$SrF_2$ ternary system consists of a continuous field of an isovalent fluorite solid solution. It is limited to two binary systems with minima on the melting curves and one with a solid solution without extremes. The assumed phase portrait (crystallization surface) of the $PbF_2$–$CdF_2$–$SrF_2$ system according to [25] is schematically shown in Figure 1 based on the reference points—the melting point of individual components and binary congruent compositions. Lines with arrows correspond to the direction of melt crystallization. One can expect the formation of "temperature troughs" on the liquidus surface along the sections indicated by dashed lines. Therefore, for the growth of crystals, compositions lying in these two sections of the ternary system were chosen. Specific compositions are indicated as circles in Figure 1 and are listed in Table 2. Obviously, the melting points of these compositions will not go beyond the range of 1023 K ($Pb_{0.67}Cd_{0.33}F_2$) to 1298 K ($Cd_{0.75}Sr_{0.25}F_2$).

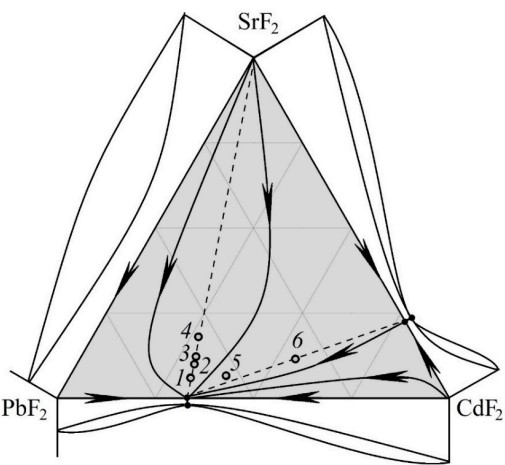

**Figure 1.** Schematic representation of the liquidus surface of the $PbF_2$–$CdF_2$–$SrF_2$ system. Arrows show the course of crystallization lines (phase portrait [27]). Dashed lines are the sections where the compositions for crystallization were selected (points 1–6).

**Table 2.** Composition, density, and refractive indices of the grown crystals.0.

| No | The Crystal Composition, mol. % | | | | | | $a$, Å | $\rho_{calc.}$, g/cm³ | $n$ ($\lambda = 632.8$ nm) |
|---|---|---|---|---|---|---|---|---|---|
| | By Charge | | | According to XRFA (Center) | | | | | |
| | $PbF_2$ | $CdF_2$ | $SrF_2$ | $PbF_2$ | $CdF_2$ | $SrF_2$ | | | |
| 1 | 63 | 31 | 6 | 62.1 | 30 | 7.9 | 5.7552(2) | 7.2212(5) | 1.7084 |
| 2 | 60 | 30 | 10 | 58.9 | 29.2 | 11.9 | 5.7606(3) | 7.0611(5) | 1.6862 |
| 3 | 58.5 | 29.3 | 12.2 | 59.1 | 27.7 | 13.2 | 5.7678(3) | 7.0301(5) | 1.6963 |
| 4 | 55 | 27 | 18 | 55.2 | 27.3 | 17.5 | 5.7717(8) | 6.8514(5) | 1.6792 |
| 5 | 53.6 | 39.8 | 6.6 | 51.6 | 39.8 | 8.5 | 5.7024(3) | 7.0566(5) | 1.6823 |
| 6 | 33.5 | 55 | 11.5 | 33 | 54.1 | 12.9 | 5.6406(2) | 6.6039(5) | 1.6403 |
| 7 | 66.7 | 33.3 | 0 | 66.7 | 33.3 | 0 | 5.7317(3) | 7.5232(5) | 1.7049 |
| 8 | 100 | 0 | 0 | 100 | 0 | 0 | 5.939(4) | 7.7706(5) | 1.7611 |
| 9 | 0 | 100 | 0 | 0 | 100 | 0 | 5.388(2) | 6.3849(5) | 1.5726 [35] |
| 10 | 0 | 0 | 100 | 0 | 0 | 100 | 5.800(2) | 4.2750(5) | 1.4371 [36] |

The growth experiment was carried out in a multi-celled crucible in one step. Accordingly, all compositions were brought to the same temperature, 1298 K. Thus, different degrees of uncontrolled melt overheating were created for the grown crystal compositions.

As a result of crystallization, transparent boules with a diameter of 12 mm and a length of up to 30 mm (Figure 2a) of six compositions ($Pb_{1-x}Cd_x$)$_{1-y}$Sr$_y$F$_2$ and reference

crystals of $PbF_2$ and $Pb_{0.67}Cd_{0.33}F_2$ were obtained. Evaporation losses were different due to different melting temperatures and vapor pressures for all compositions but did not exceed 5 wt.%.

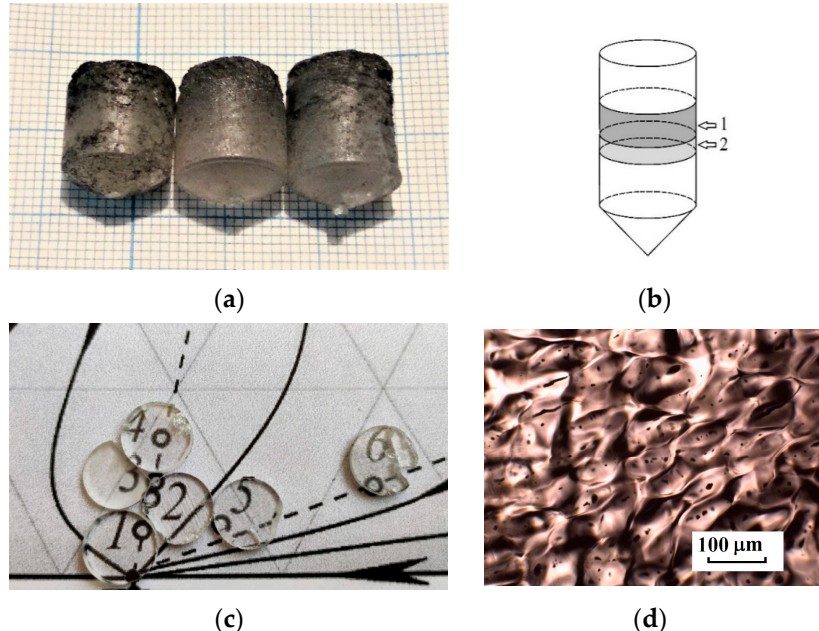

**Figure 2.** External view of some crystal boules "as grown" (**a**), scheme of boule preparation: 1—slice for non-destructive studies, 2—slice for XRD analysis (**b**), appearance of polished disks (**c**), cellular substructure in the $Pb_{0.33}Cd_{0.541}Sr_{0.129}F_2$—sample 6 (optical photography through transmission) (**d**).

Slices 2–4 mm thick were cut from the middle of the grown crystals (Figure 2b) and polished for optical studies. The average composition of these samples, determined using XRFA from four points on the disc surface, is shown in Table 2. Despite the low accuracy of the XRFA, one can state a slight difference in the compositions of the central parts of the crystals from the bulk composition of the mixture.

The main mechanism causing the inhomogeneity of the crystal composition during directional crystallization of the melt is its concentration supercooling, which leads to the loss of stability of the crystallization front (or morphological instability) [37]. For the directional crystallization of a two-component solid solution, the criterion for the crystallization front stability with respect to concentration supercooling is given by the following expression:

$$GD/v > m\Delta x, \tag{2}$$

where $v$ is the crystallization rate, $G$ is the temperature gradient at the crystallization front, $D$ is the interdiffusion coefficient of the components in the melt, $m$ is the liquidus line slope, and $\Delta x = x_S - x_L$ is the concentration jump at the crystallization front.

Expression (2) is a generalized version of the well-known Tiller criterion. The right side of the expression, called the stability function $F = m\Delta x$ [38], can be easily determined as a first approximation from the equilibrium phase diagram of the binary system. The calculation of dependences $F(x)$ allows one to select technological parameters (crystallization rate $v$ and temperature gradient $G$) for growing non-cellular two-component crystals so that the value of $GD/v$ exceeds the calculated critical value $F(x)$ [39,40]. In this case, it should be taken into account that the reduction in the rate of the crucible and the creation of high-gradient thermal conditions at the crystallization front are also limited by technological capabilities. With three or more components, the introduction of a stability function by analogy with two-component systems is impossible [27], so the search for crystallization conditions and regions of homogeneous cell-free crystal growth remains a complex experimental task.

Polished optical elements cut from the central crystal parts were examined using an optical microscope. The cellular inhomogeneity of crystals in the $PbF_2$–$CdF_2$–$SrF_2$ system is enhanced for a number of samples: 1→5, 2→3→4→6 (Figure 2c). In the vicinity of the temperature minimum point of the $PbF_2$–$CdF_2$ system, there is a region of cell-free growth of $(Pb_{0.67}Cd_{0.33})_{1-y}Sr_yF_2$ ($0 < y < 0.05$) (at a crystallization rate of up to 6 mm/h). As the concentration of $SrF_2$ increases, the inhomogeneity increases. Figure 2d shows a photograph of sample 6, which demonstrates a highly developed cellular substructure.

XRD analysis was performed on the powder from the region adjacent to the disk under study (see Figure 2b). As expected, the XRD patterns show a single phase, which is identified as a solid solution with a fluorite-type structure (sp. gr. *Fm-3m*).

The unit cell parameters for all samples are given in Table 2. Figure 3 shows, as an example, XRD patterns of two crystal samples: visually homogeneous $Pb_{0.621}Cd_{0.30}Sr_{0.079}F_2$ and highly cellular $Pb_{0.33}Cd_{0.541}Sr_{0.129}F_2$. The broadening of the reflections and the unresolved $K\alpha 1/\alpha 2$ doublets were observed due to the inhomogeneity of the last composition.

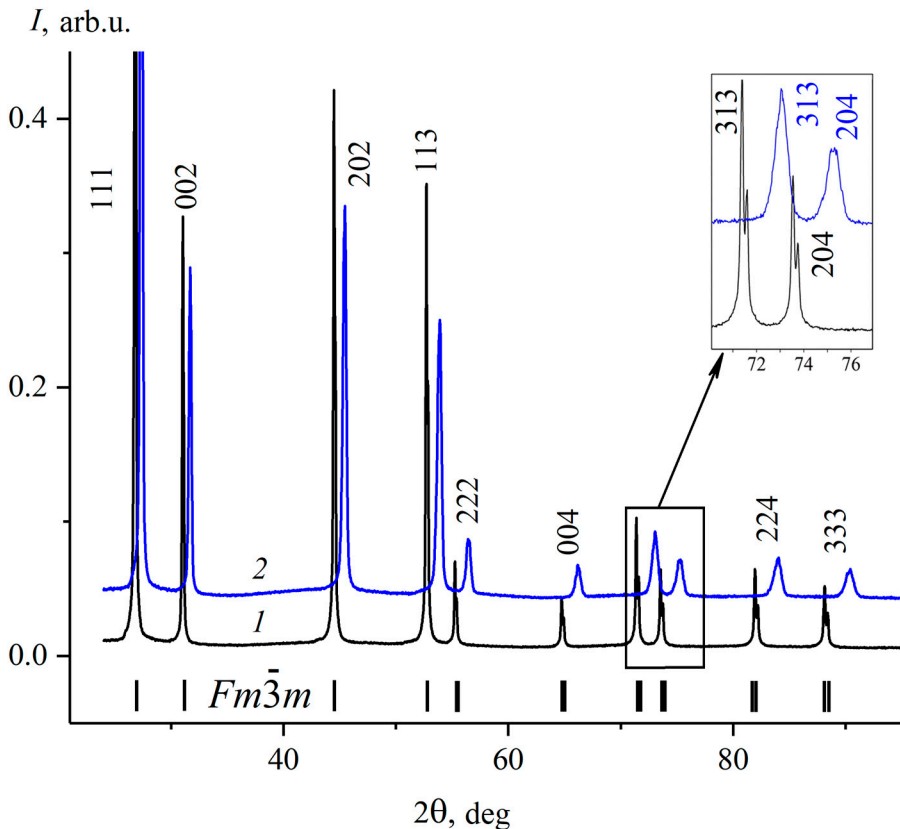

**Figure 3.** XRD patterns of $Pb_{0.621}Cd_{0.30}Sr_{0.079}F_2$ (1) and $Pb_{0.33}Cd_{0.541}Sr_{0.129}F_2$ (2). The dashes show the positions of the Bragg reflections for sp. gr. *Fm-3m* with lattice parameter $a = 5.7552(2)$ Å.

Since the $PbF_2$, $CdF_2$, and $SrF_2$ components have different vapor pressures and distribution coefficients different from unity, then, as a rule, the composition of the grown ternary crystals differs greatly from the composition of the bulk charge. To create materials with specified predictable properties, knowledge of empirical property relationships is essential, for example, the composition–density relationship or the composition–refractive index relationship. If we accept the position that the unit cell parameters of binary solid solutions in all three systems are additive (subject to Vegard's rule), then they can be described by the following equations:

$$a_{Pb\text{-}Cd} = a_{Pb} + \kappa_1 x, \quad a_{Pb\text{-}Sr} = a_{Pb} + \kappa_2 y, \quad a_{Cd\text{-}Sr} = a_{Cd} + \kappa_3 y, \tag{3}$$

where the coefficients are $\kappa_1 = a_{Pb} - a_{Cd}$, $\kappa_2 = a_{Pb} - a_{Sr}$, and $\kappa_3 = a_{Cd} - a_{Sr}$, and $a_{Pb} = 5.939(4)$, $a_{Cd} = 5.388(2)$, and $a_{Sr} = 5.800(2)$Å are the unit cell parameters of pure components—lead, cadmium, and strontium fluorides, respectively. (The form of writing the formula of a ternary solid solution is $Pb_{1-x-y}Cd_xSr_yF_2$.)

Substituting the values, we obtain an analytical expression for the concentration dependence of the unit cell parameter for a ternary solid solution:

$$a_{Pb\text{-}Cd\text{-}Sr} = a_{Pb} - (a_{Pb} - a_{Cd})x - (a_{Pb} - a_{Sr})y = 5.939 - 0.551x - 0.139y, \qquad (4)$$

where $x$ and $y$ are the concentrations of $CdF_2$ and $SrF_2$, respectively. For clarity, the plane of unit cell parameters is shown in Figure 4a. The light blue spheres show the values calculated by Equation (4) (they lie in the plane), and the blue spheres show the experimental values of the unit cell parameters. It can be seen that for the $Pb_{0.67}Cd_{0.33}F_2$ sample, some negative deviation from the Vegard rule was recorded. For a ternary solid solution, the values have both negative and positive deviations. On the whole, the $a_{Pb–Cd–Sr}(x, y)$ dependence is satisfactorily described by the plane equation.

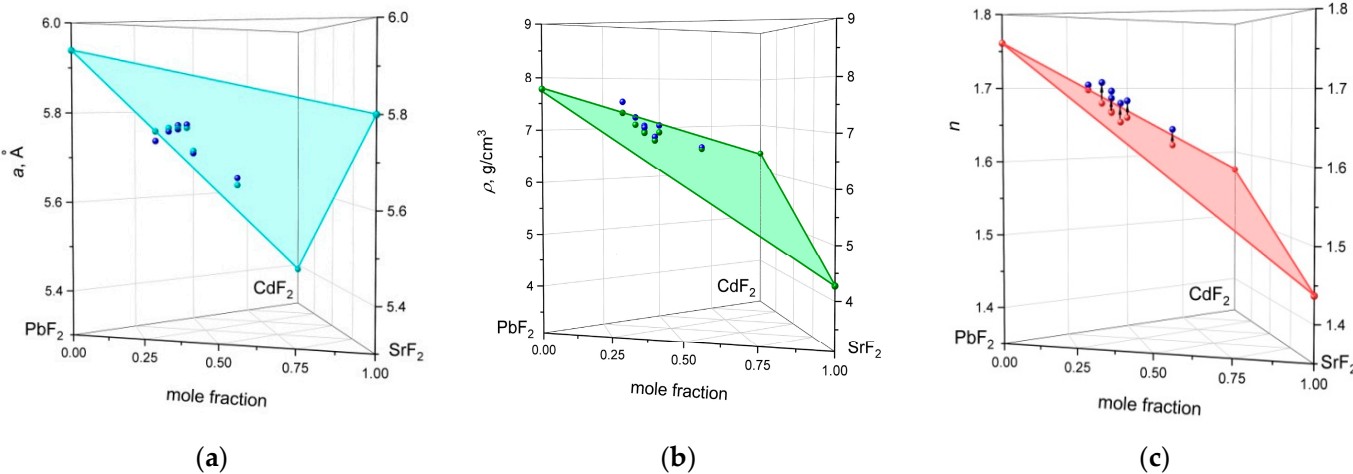

**Figure 4.** Concentration dependences of the unit cell parameters $a$ (**a**), X-ray densities $\rho$ (**b**), and refractive indexes $n$ (**c**) in the ternary system. The light blue, green, and red spheres in the corresponding figures were calculated by additivity (they lie in the plane), and the blue spheres are the experimental values.

Similarly, we can derive the dependence of the solid solution density on the composition $\rho_{Pb–Cd–Sr}(x, y)$:

$$\rho_{Pb\text{-}Cd\text{-}Sr} = \rho_{Pb} - (\rho_{Pb} - \rho_{Cd})x - (\rho_{Pb} - \rho_{Sr})y = 7.7706 - 1.3857x - 3.4956y. \qquad (5)$$

The X-ray density values for the samples under study are given in Table 2; the density plane is shown in Figure 4b.

$Pb_{1-x-y}Cd_xSr_yF_2$ solid solutions can provide optical materials with a refractive index $n_D$ ranging from 1.4371 (for pure $SrF_2$) to 1.7611 (for pure $PbF_2$), i.e., they provide a variation in the refractive index over a very wide range. Similarly, under the condition of additivity, we obtain an analytical expression for the concentration dependence of the refractive index:

$$n_{Pb\text{-}Cd\text{-}Sr} = n_{Pb} - (n_{Pb} - n_{Cd})x - (n_{Pb} - n_{Sr})y = 1.4371 + 0.3240x + 0.1355y, \qquad (6)$$

Where $n_{Pb}$, $n_{Cd}$, and $n_{Sr}$ are the refractive indices of lead, cadmium, and strontium fluorides, respectively (see Table 2). This plane is shown in Figure 4c (red spheres are values calculated from additivity; blue spheres are measured values). It can be seen that there is a significant positive deviation from additivity in the system. Nevertheless, concentration dependences (4)–(6) can be used for an approximate assessment of the material composition

at known (measured) *a* or ρ and *n* or, conversely, to predict the crystal properties of a given composition of *x* and *y*.

Thus, Equations (4)–(6) and Figure 4a–c clearly illustrate the possibility of varying the density and refractive index of the ternary solid solution in a wide range: from 4.27 g/cm$^3$ and 1.4371 (for SrF$_2$) to 7.77 g/cm$^3$ and 1.7611 (for PbF$_2$), respectively. The above analysis shows a potential approach to predicting the composition of a solid solution with the required practically significant physical properties.

### 3.2. Optical Properties

The transmission spectra of crystalline samples of Pb$_{0.621}$Cd$_{0.30}$Sr$_{0.079}$F$_2$, Pb$_{0.33}$Cd$_{0.541}$Sr$_{0.129}$F$_2$, and PbF$_2$ (for comparison) are shown in Figure 5. The PbF$_2$ crystal is transparent in the range from 0.25 to 15 μm, which corresponds to the literature data [41,42]. The low level of transmission is associated with the optical processing of samples under conditions that are not optimal for hydrolysable materials. The optimization of the optical polishing process for these crystals is in progress. The additional observed absorption in the region of 0.3 μm is associated with the uncontrolled content of Ce$^{3+}$ ions [43].

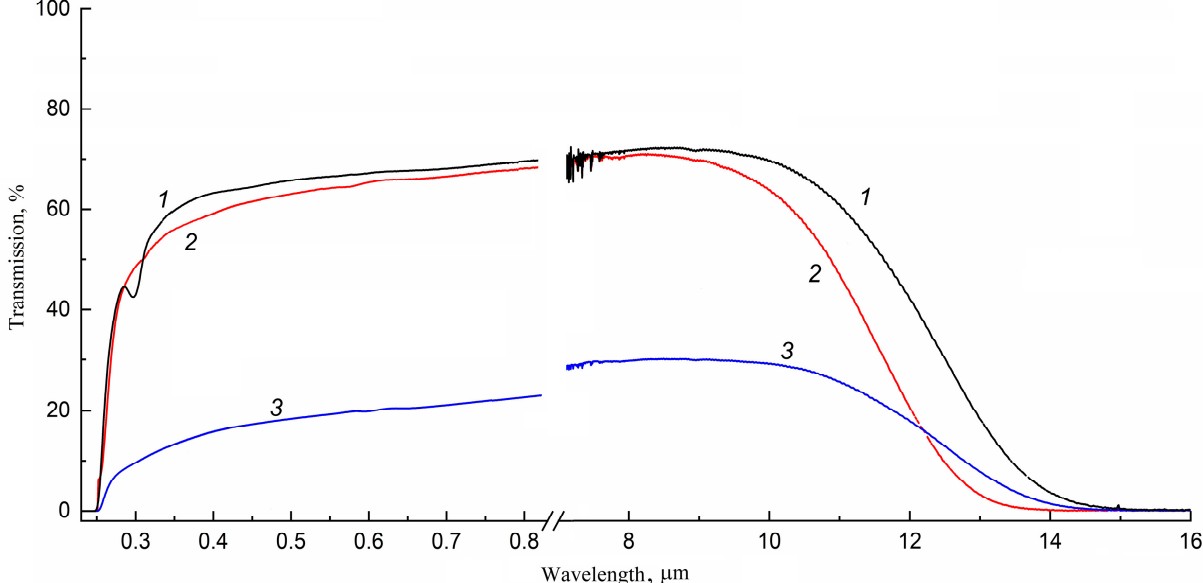

**Figure 5.** Transmission spectra of the studied crystals: PbF$_2$ (1), Pb$_{0.621}$Cd$_{0.30}$Sr$_{0.079}$F$_2$ (2), and Pb$_{0.33}$Cd$_{0.541}$Sr$_{0.129}$F$_2$ (3). The thickness of the samples is 2 mm.

The presence of CdF$_2$ in the composition of multicomponent matrices leads to an insignificant red shift of the short-wavelength transmission cut-off of these Cd-containing crystals [34]. The IR transmission cut-off of ternary crystals also regularly shifts to the short-wavelength side. Strongly inhomogeneous cellular crystals retain transparency in a wide range at the level of 20% (spectrum *3*).

### 3.3. Ionic Conductivity

Next, the electrically conductive properties of the Pb$_{0.552}$Cd$_{0.273}$Sr$_{0.175}$F$_2$ sample were studied, which can be represented as a Pb$_{0.67}$Cd$_{0.33}$F$_2$ matrix doped with SrF$_2$—(Pb$_{0.67}$Cd$_{0.33}$)$_{1-y}$Sr$_y$F$_2$ (*y* = 0.175). The impedance hodograph $Z^*(\omega) = Z'(\omega) + iZ''(\omega)$ of the electrochemical system Ag|crystal|Ag and its equivalent electrical circuit are presented in Figure 6a and 6b, respectively. Here, the real and imaginary parts of the complex impedance are $Z' = |Z| \times \text{Cos } \varphi$ and $Z'' = |Z| \times \text{Sin } \varphi$, respectively, $|Z|$ is the impedance modulus, φ is the phase angle between voltage and current, and ω is the circular frequency.

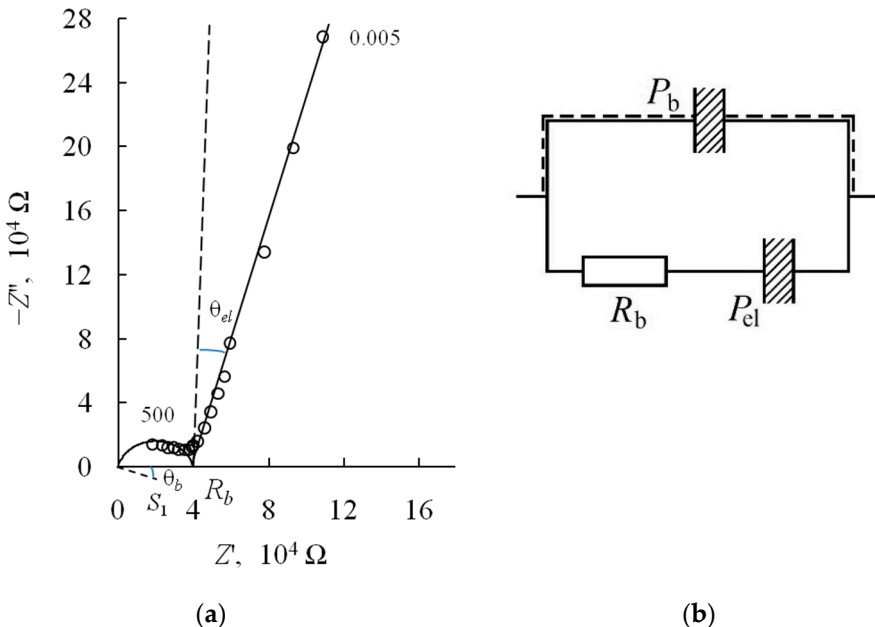

**Figure 6.** Hodograph of the impedance $Z^*(\omega) = Z' + iZ''$ of the system Ag | Pb$_{0.552}$Cd$_{0.273}$Sr$_{0.175}$F$_2$ | Ag at 293 K (**a**) and equivalent circuitry simulating impedance (**b**). Numerals are electric field frequency in kHz; S$_1$ is the center of the circle approximating the measurement results.

In an equivalent AC circuit, the resistor $R_b$ determines the bulk resistance of the crystal associated with the process of electrotransfer, $P_b(\omega)$ and $P_{el}(\omega)$ frequency-dependent elements with a constant phase angle (constant phase elements, CPE) [34], model polarization processes in the bulk of the crystal, and the electrode/crystal interface. The admittance $Y^*(\omega) = 1/Z^*(\omega)$ of $P_b(\omega)$ and $P_{el}(\omega)$ CPE elements can be represented as $Y_P^* = Y_0(i\omega)^n$, $0 \leq n \leq 1$. For $n = 1$, the elements $P_b$ and $P_{el}$ turn into geometric capacitance $C_g$ and double layer capacitance $C_{dl}$, respectively. The parameters of the impedance spectrum were determined with the nonlinear squares method using the FIRDAC software package [44]. The equivalent circuit parameters are $R_b = (3.95 \pm 0.09) \times 10^4$ $\Omega$, $Y_{0,b} = (5.9 \pm 2.2) \times 10^{-10}$ S $\times$ (Hz)$^{-n}{}_b$, $n_b = 0.72 \pm 0.06$, $Y_{0,el} = (1.09 \pm 0.08) \times 10^{-8}$ S $\times$ (Hz)$^{-n}{}_{el}$, and $n_{el} = 0.81 \pm 0.04$. In Figure 6, the calculated depression angles $\theta = \pi(1 - n)/2$ are $\theta_b = 24.9°$ and $\theta_{el} = 17.5°$. Large values of the depression angles indicate the inhomogeneity of electrical processes in the bulk of the crystal and at the electrode/crystal interface, which is due to the structural inhomogeneity of the sample under study.

Impedance measurements made it possible to reliably determine the bulk resistance $R_b$ of a Pb$_{0.552}$Cd$_{0.273}$Sr$_{0.175}$F$_2$ single crystal, which corresponds to the intersection of the $Z^*(\omega)$ impedance hodographs of the electrochemical cell with the $Z'$ axis of active resistances or the fulfillment of the condition: the phase shift angle between voltage and current $\varphi = 0°$. The calculated value of the direct current electrical conductivity of the crystal is $\sigma_{dc} = h/(SR_b) = 1.5 \times 10^{-5}$ S/cm at 293 K, where the geometric factor $h/S = 0.6$ cm$^{-1}$, $h$ is the sample thickness, and $S$ is the electrode area.

The presence of the capacitive effect from inert (silver) electrodes at low frequencies in the impedance spectrum (Figure 6b) indicates the ionic nature of the electrotransport. The nature of the ionic conductivity of the fluorite-type solid solution Pb$_{0.552}$Cd$_{0.273}$Sr$_{0.175}$F$_2$ is due to fluorine anions. This is directly indicated by the results of the F$^{19}$ NMR study of Pb$_{0.67}$Cd$_{0.33}$F$_2$ crystals, in which a high diffusion of F$^-$ ions was found [45,46], and theoretical calculations using molecular dynamics and quantum chemistry methods [47,48].

Figure 7 shows the temperature dependence of the ionic conductivity for the (Pb$_{0.67}$Cd$_{0.33}$)$_{0.825}$Sr$_{0.175}$F$_2$ crystal in the temperature range of 293–406 K.

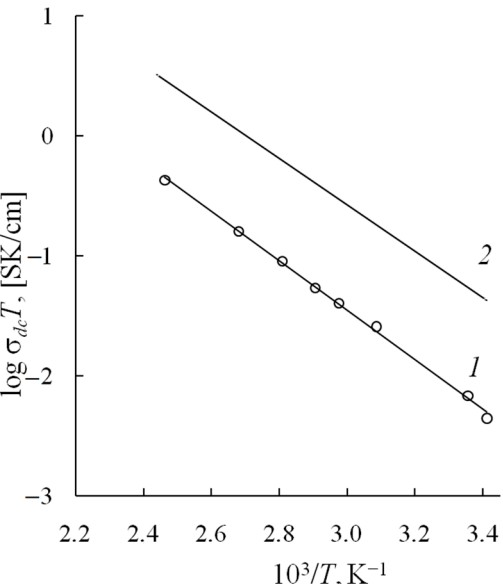

**Figure 7.** Temperature dependence of the ionic conductivity $\sigma_{dc}(T)$ for single crystals: $(Pb_{0.67}Cd_{0.33})_{0.825}Sr_{0.175}F_2$ (1) and $Pb_{0.67}Cd_{0.33}F_2$ [30,32] (2).

The conductivity increases by a factor of 70 with a rise in temperature range from 293 to 406 K. The electrophysical data were processed with the Arrhenius-type equation:

$$\sigma_{dc}T = A\exp(-H_\sigma/kT),$$

where $A$ is the pre-exponential factor, and $H_\sigma$ is the electric transfer activation enthalpy. The parameters of the Arrhenius equation are as follows: $A = 5.3 \times 10^4$ SK/cm and $H_\sigma = 0.408 \pm 0.005$ eV.

The comparison of the ionic conductivity of the $Pb_{0.552}Cd_{0.273}Sr_{0.175}F_2 = (Pb_{0.67}Cd_{0.33})_{0.825}Sr_{0.175}F_2$ ternary single crystals and the $Pb_{0.67}Cd_{0.33}F_2$ two-component crystals ($\sim 2 \times 10^{-4}$ S/cm) [29,32] shows that the doping of $Pb_{0.67}Cd_{0.33}F_2$ with $Sr^{2+}$ cations leads to a decrease in the ionic conductivity by an order of magnitude (Figure 7). The ionic conductivity activation enthalpy for the $(Pb_{0.67}Cd_{0.33})_{0.825}Sr_{0.175}F_2$ crystal is bigger than the corresponding value for the $Pb_{0.67}Cd_{0.33}F_2$ crystal ($H_\sigma = 0.36$–$0.38$ eV [30,32]). The increase in the potential barrier for the migration of fluorine ions ($H_\sigma$ value) leads to the decrease in the ionic conductivity when $Sr^{2+}$ cations are introduced into the $Pb_{0.67}Cd_{0.33}F_2$ fluorite-type structure.

The ionic radii of the $Pb^{2+}$, $Cd^{2+}$, and $Sr^{2+}$, cations included in the ternary solid solution are 0.143, 0.124, and 0.140 nm, respectively [49]. It can be assumed that the $Sr^{2+}$ cations will replace $Pb^{2+}$ cations in the crystal lattice. Therefore, the reason for the decrease in conductivity of the $Pb_{0.552}Cd_{0.273}Sr_{0.175}F_2$ solid solution is apparently the deviation of the cadmium content (27 mol.% $CdF_2$) from the optimal content (33 mol.% $CdF_2$), realized in the $Pb_{0.67}Cd_{0.33}F_2$ solid solution. However, the ternary solid solution can also be attributed to low-temperature ionic conductors. Its conductivity is slightly lower compared to that of the dual composition, but the doping of the composition with $SrF_2$ allows you to stabilize the multicomponent material in terms of melting point.

## 4. Conclusions

Three-component crystals of the $Pb_{1-x-y}Cd_xSr_yF_2$ solid solution with a fluorite structure were grown using a directional crystallization technique from the melt. The region of cell-free growth was experimentally determined. Crystals of compositions from this region were characterized—the unit cell parameter (and, accordingly, the density), refractive index, transmission spectra, and electrochemical impedance were analyzed. Homogeneous crystals can be recommended as optical structural materials and hosts for doping with

active REE ions due to the high isomorphic capacity of the fluorite structure. Cellular crystals can be used to produce ceramics using the hot forming technique [50] and low-temperature ionic conductors.

The present study opens the way to the creation of new promising multicomponent materials with controlled fundamental properties.

The synthesis and research of four- and more component single crystals based on alkaline earth metal fluorides is planned to develop an understanding of the possibilities of further complicating the quantitative and qualitative composition.

**Author Contributions:** Conceptualization, XRD, calculations, and crystal growth, I.I.B.; optical investigation, I.O.G., D.N.K., and V.I.S.; ionic conductivity investigation, N.I.S.; writing, visualization, and editing, all authors. All authors have read and agreed to the published version of the manuscript.

**Funding:** This research was performed within the State assignment of Federal Scientific Research Center "Crystallography and Photonics" of the Russian Academy of Sciences using the equipment of the Shared Equipment Center of the Federal Scientific Research Centre "Crystallography and Photonics" of the Russian Academy of Sciences.

**Data Availability Statement:** Not applicable.

**Acknowledgments:** We are grateful to A.G. Saveliev and B.V. Nabatov for their help in performing measurements and P.P. Fedorov for the inspiration for this research.

**Conflicts of Interest:** The authors declare no conflict of interest.

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
