# Peer review of "PbF2–CdF2–SrF2 Ternary Solid Solution: Crystal Growth and Investigation"

_condensedmatter, doi:10.3390/condmat8030073_

Round 1
Reviewer 1 Report
The manuscript is of high quality and it presents very well all the results of investigation on PbF2-CdF2-SrF2 ternary solid solutions with a fluorite structure, which were grown from melt by the vertical directional crystallization technique.
I suggest that the manuscript be accepted for publication after correcting the few typos I have noticed:
- line 39: replace "и" with "and";
- line 66: replace "Ca0.3(3)Sr0.3(3)Sr0.3(3)F2" with "Ca0.3(3)Sr0.3(3)Ba0.3(3)F2";
- line 71: replace "The ternary system PbF2-CdF2-SrF2 is system is attracrive..." with "The ternary system PbF2-CdF2-SrF2 is system that is attractive..."
Minor editing of English language required.
Author Response
Thank you very much for your careful reading of our manuscript! We have taken into account the above remarks (changes are highlighted in yellow in the text).
Reviewer 2 Report
Buchinskaya et al. reported the crystal growth and property characterization results of Pb0.621Cd0.30Sr0.079F2 and Pb0.33Cd0.541Sr0.129F2 compounds from PbF2-CdF2-SrF2 system. From the first glance, this work is indeed warranted for publication in the journal Condensed Matter. However, after further reading, I find that the current version has many parts that are unclear and needs further improvement. The work itself is scientifically interesting, yet the discussion and formatting is far from publication. Below I am giving a few examples and based on above, I am not able to recommend the current version for publishing.
1. Have authors done any single crystal XRD on these compounds?
2. I noticed authors have cited many of their own publications in the reference section. Authors should address the concern and clarify if indeed all those citations are necessary.
3. Caption of Table 2 can be re-phrased. ‘Some parameters’ is not clear.
4. Have Fig. 3 been discussed in the main text? I could not locate any discussions for Fig. 3 XRD pattern. Besides, what is the unit grad in x-axis? To the best of my understanding, this is usually labelled as 2-theta, deg. What is the reason for peak shifting e.g., (002), (113), (222)?
5. Fig.5, authors should change the x-axis to either wavelength or photon energy, so readers can know the bandgap energy for these compounds at their first glance. See https://doi.org/10.1016/S0022-0248(02)01579-8 (reference [40], cited by authors)
6. Caption of Fig. 2, I do not understand. What does author mean ‘XRD Appearance of some crystal boules…’? What is ‘XRD Appearance’? Also, Fig. 2d has not been discussed in the main text and what is Fig. 2d?
Moderate English language editing and check is required.
Author Response
We thank the Reviewer for careful consideration of our manuscript and for useful comments.
Comment 1
Have authors done any single crystal XRD on these compounds?
Response 1
No, they didn't. In general, we have such an opportunity. But this is a separate serious work together with other specialists. In this work, there is no need for single-crystal analysis, the task was to obtain an optically transparent crystal, to determine its phase composition and estimate the density can be easily using powder XRD.
Comment 2
I noticed authors have cited many of their own publications in the reference section. Authors should address the concern and clarify if indeed all those citations are necessary.
Response 2
We have revised the references. As a result, we have replaced several self-citations with others on the same topic.
Comment 3
Caption of Table 2 can be re-phrased. ‘Some parameters’ is not clear.
Response 3
Agree. Done in the attached text.
Comment 4
Have Fig. 3 been discussed in the main text? I could not locate any discussions for Fig. 3 XRD pattern. Besides, what is the unit grad in x-axis? To the best of my understanding, this is usually labelled as 2-theta, deg. What is the reason for peak shifting e.g., (002), (113), (222)?
Response 4
Description Fig. 3 has been added to the text. X-ray patterns belong to solid solutions of different compositions (with different lattice parameters), so the shift of reflections is a matter of course. Main task Fig. 3 - illustrate the shape of reflections responsible for crystallinity.
Comment 5
Fig.5, authors should change the x-axis to either wavelength or photon energy, so readers can know the bandgap energy for these compounds at their first glance. See https://doi.org/10.1016/S0022-0248(02)01579-8 (reference [40], cited by authors).
Response 5
The x-axis in our figure shows the wavelength λ, µm. We designed the graph by analogy with [40 → 41].
Comment 6
Caption of Fig. 2, I do not understand. What does author mean ‘XRD Appearance of some crystal boules…’? What is ‘XRD Appearance’? Also, Fig. 2d has not been discussed in the main text and what is Fig. 2d?
Response 6
It's a typo! "XRD" was removed, the name was changed. Additional information has been added.
Reviewer 3 Report
The manuscript "PbF2-CdF2-SrF2 ternary solid solution: crystal growth and investigation" by Irina I. Buchinskaya, Ivan O. Goryachuk, Nikolay I. Sorokin, Victor I. Sokolov, Denis N. Karimov is a complete study of the growth of Pb1 - x - уСdхSrуF2 (0.27 < 9 x < 0.55, 0.06 < y < 0.18) solid solution crystals in a ternary fluoride system and the study of their optical properties and ionic conductivity.
The paper is worthy of publication in a Condensed Matter journal. However, I have the following questions
1) Figure 1. This schematic diagram is completely impossible to read. For points 1-8, it is not clear to what temperatures the fluoride mixtures were heated. Please provide a table in the text showing the temperatures to which the fluoride powders were heated in each of cases 1-8.
2) I would like to see the Arrhenius dependence of the conductivity for a single crystal grown in the ternary system Pb0.552Сd0.273Sr0.175F2 and for the initial single crystal Pb0.67Cd0.33F2 not doped with strontium. Please, provide an additional figure with Arrhenius curves and discuss them in detail.
3) The authors have published a significant number of papers on fluoride systems. Please, give examples of the practical use of such materials in optics or as ionic conductors.
English level is quite acceptable.
Author Response
Comment 1
Figure 1. This schematic diagram is completely impossible to read. For points 1-8, it is not clear to what temperatures the fluoride mixtures were heated. Please provide a table in the text showing the temperatures to which the fluoride powders were heated in each of cases 1-8.
Response 1
Thank you very much! This is a very important note; indeed, we did not write anything about the melting temperatures of the samples, only casually mentioned evaporation losses in the lines.
Information was added to the text of the manuscript explaining the choice of compositions and Fig. 1.
Growing crystals of inorganic fluorides is a complex process that requires a special high-temperature furnace with an aggressive fluorinating atmosphere, in addition, we work with highly volatile PbF2 and CdF2 components. Therefore, the experiment to obtain crystals was carried out in 1 stage in a multi-celled crucible. Accordingly, all compositions were brought to the same temperature - 1025°C. It would be possible to take the compositions lying along the sections closer to SrF2 (Tmelt=1473°C), but then the evaporation losses of the most fusible compositions would be too large. Therefore, we limited ourselves to compositions 1-6, indicated in Fig. 1. (Moreover, the most homogeneous area - "temperature troughs" - is expected near Pb0.67Cd0.33F2).
Comment 2
I would like to see the Arrhenius dependence of the conductivity for a single crystal grown in the ternary system Pb0.552Сd0.273Sr0.175F2 and for the initial single crystal Pb0.67Cd0.33F2 not doped with strontium. Please, provide an additional figure with Arrhenius curves and discuss them in detail.
Response 2
Added to the manuscript are Fig. 7 and its discussions.
Comment 3
The authors have published a significant number of papers on fluoride systems. Please, give examples of the practical use of such materials in optics or as ionic conductors.
Response 3
Difluorides of alkaline earth metals, both without impurities and doped with active REE ions, are used as scintillator crystals for Positron Emission Tomography (PET), materials for EM calorimeters (https://www.msesupplies.com/products/lead-fluoride-pbf-2-single-crystal); film structures - optical coatings, and others. Fluorine ion battery and electrode materials are a separate issue.
Kosacki, I. Physical properties and applications of Cd1-xPbxF2 superionic crystals. Appl. Phys. A. 1989, 49, 413-424. https://doi.org/10.1007/BF00615026
Groult H., Tressaud A. Use of inorganic fluorinated materials in lithium batteries and in energy conversion systems. Chem. Commun. 2018; 54: 11375-11382. https://doi.org/10.1039/C8CC05549A
Weller P. F. Electrical and Optical Studies of Doped CdF2-CaF2 Crystals. Inorg. Chem. 1966, 5, 736–739. https://doi.org/10.1021/ic50039a009
Lui, M., McFarlane, R.A. & Yap, D. Growth of Erbium Doped PbF2-SrF2 Epitaxial Layers On GaAs(111)B for Upconversion Waveguide Laser Applications. MRS Online Proceedings Library. 1993, 329, 167–172. https://doi.org/10.1557/PROC-329-167
Ntoupis, V.; Linardatos, D.; Saatsakis, G.; Kalyvas, N.; Bakas, A.; Fountos, G.; Kandarakis, I.; Michail, C.; Valais, I. Response of Lead Fluoride (PbF2) Crystal under X-ray and Gamma Ray Radiation. Photonics 2023, 10, 57. https://doi.org/10.3390/photonics10010057
Yang, W.Bi, X. Li, M. Liao, W. Gao, Y. Ohishi, Y. Fang, and Y. Li. Ultrabroadband supercontinuum generation through filamentation in a lead fluoride crystal. J. OSA B. 2019, 36(2), A1-7.
Spilker, P.L. Cole, P. Bertin, T.A. Forest, M. Mestari, S. Naeem, J. Roche, C. Munoz, and N. LeBaron. Optical Restoration of Lead Fluoride Crystals. AIP Conference Proceedings. 2009, 1099, 997. https://doi.org/10.1063/1.3120211
Mao, L. Zhang and R. Zhu. A Search for Scintillation in Doped Cubic Lead Fluoride Crystals. IEEE Transactions on Nuclear Science. 2010, 57(6), 3841-3845. https://doi:10.1109/TNS.2010.2076372
We have added several such references to our manuscript.
Reviewer 4 Report
I have read the paper, and my general opinion about this work is positive. Growing new materials is an essential and not easy task. The paper's general quality is acceptable, and all the analysis seems correct. The authors analyzed the properties of the obtained material for different compositions. The research is detailed and contains valuable results. The paper is clearly written and well organized. I think the language and style are OK, but still, there are some mistakes or typo errors. Here are some more comments on this paper:
- - In the Introduction, line 71, there is: The ternary system PbF2–CdF2–SrF2 is system is…should be just is attractive.
- - The growing temperature should be provided for all crystals since it is a crucial parameter. Also, I would expect more details here concerning growing, such as powder preparation, purification, and after-growing sample preparation. Some materials are hygroscopic (for sure, SrF2) and require special treatment. So the question is whether were powder dried before weighing and so on.
- - The same question arises for grown crystals. They are hygroscopic. Dealing with such materials, measurements, and sample preparation is not easy. For the same reason, the application of such materials is limited. In such a case, how were the measurements conducted? Also, did the authors check the grown crystals' stability in the air?
- - All formulae concerning the Vegards rule should be just written and numbered as equations
- - I have found an ethical issue, that is inappropriate self-citations:
Primary author Buchinskaya 16 times
Sorokin 9 times
Karimov 7 times,
All together, almost half of all references! I would expect more citations from other authors working in this field. The general scientific quality is not bad, but the authors should reduce self-citations significantly.
In summary, I recommend a major revision of the paper.
Author Response
We would like to express our gratitude to the reviewer for checking the manuscript and making minor comments that will help make it clearer for readers and more visually perfect for perception.
Comment 1
- In the Introduction, line 71, there is: The ternary system PbF2–CdF2–SrF2 is system is…should be just is attractive.
Response 1
We are grateful for the finding a typos and suggestions for improvement. These remarks have been corrected in the text.
Comment 2
- The growing temperature should be provided for all crystals since it is a crucial parameter. Also, I would expect more details here concerning growing, such as powder preparation, purification, and after-growing sample preparation. Some materials are hygroscopic (for sure, SrF2) and require special treatment. So the question is whether were powder dried before weighing and so on.
Response 2
We have added the details of the growth experiment to the text. All powders were carefully dried in a vacuum and remelted in a fluorinating atmosphere to obtain transparent polycrystalline compacts. And only then they were used in growth experiments.
Comment 3
- The same question arises for grown crystals. They are hygroscopic. Dealing with such materials, measurements, and sample preparation is not easy. For the same reason, the application of such materials is limited. In such a case, how were the measurements conducted? Also, did the authors check the grown crystals' stability in the air?
Response 3
These multi-component crystals are stable during storage under normal conditions. Some problems associated with hygroscopicity were manifested during optical polishing of samples using water, the surface was covered with a layer of hydroxide film. The use of acetone for polishing solves this problem. All measurements were carried out under normal room conditions. Heating of crystals during electrophysical experiments was carried out in vacuum.
Comment 4
All formulae concerning the Vegards rule should be just written and numbered as equations
Response 4
Note taken into account.
Comment 5
I have found an ethical issue, that is inappropriate self-citations:
Primary author Buchinskaya 16 times
Sorokin 9 times
Karimov 7 times,
All together, almost half of all references! I would expect more citations from other authors working in this field. The general scientific quality is not bad, but the authors should reduce self-citations significantly.
Response 5
There are 50 references in our manuscript, 19 of them are self-citations. This is 38%. Although this is not half, but definitely too much. We've shortened self-quoting and applied other references.
(This work is of a generalizing review nature. And many crystals were previously studied only by the authors of this article. In addition, the authors studied for the first time the features of crystallization of multicomponent fluoride melts.)
Round 2
Reviewer 2 Report
I think authors have given satisfactory answers for most of the questions raised. However, I am not sure if the current form can be published without ethical concern (i.e., in my opinion, this work has too many self citations). This is typically a minor issue, yet I think this is a major concern for this work. Besides, authors should also consider addressing the minor concern below.
1. Line 254, 'crystal samples are shown in Fig. 5a', which one is Fig. 5a? Unless I missed, yet there is no 'a' and 'b' labelling for Fig. 5. I also do not understand the difference between the two plots in Fig. 5. It seems one is absorbance and one is transmittance, yet they are both labelled as transmission.
Minor editing of English language is required.
Author Response
Dear reviewer,
We have fulfilled all your requirements and wishes. Thank you for your attention to our manuscript. New changes are highlighted in magenta in the text.
Comment 1
I think authors have given satisfactory answers for most of the questions raised. However, I am not sure if the current form can be published without ethical concern (i.e., in my opinion, this work has too many self citations). This is typically a minor issue, yet I think this is a major concern for this work. Besides, authors should also consider addressing the minor concern below.
Response 1
We left only 10 self-citations. This is 20% of the total number of references and is acceptable.
Comment 2
Line 254, 'crystal samples are shown in Fig. 5a', which one is Fig. 5a? Unless I missed, yet there is no 'a' and 'b' labelling for Fig. 5.
Response 2
Line 254: typo corrected (Fig. 5a→ Fig. 5).
The text says «The transmission spectra … are shown in Fig. 5», not «crystal samples are shown in Fig. 5a». The sentence has been reformulated. Other minor English corrections have also been made.
Comment 3
I also do not understand the difference between the two plots in Fig. 5. It seems one is absorbance and one is transmittance, yet they are both labelled as transmission.
Response 3
The Fig. 5 has been changed for better perception. A single discontinuous axis is drawn for the entire wavelength range.

Reviewer 4 Report
I am satisfied with the author's responses, so I believe the paper can now be accepted.
Author Response
Dear Reviewer,
Thank you for participating in our manuscript.
Round 3
Reviewer 2 Report
I now can understand what this Fig. 5 means. A better way to make this easier to follow is to revise the cation. So if authors go this way, then it will be easier to see that they are both transmission but measured at different wavelength range.
Authors have given satisfactory answers for all the concerns. Thus, I suggest an acceptance of this work for publication.